# Patterns of Spatial Variation in Rumen Microbiology, Histomorphology, and Fermentation Parameters in Tarim wapiti (*Cervus elaphus yarkandensis*)

**DOI:** 10.3390/microorganisms12010216

**Published:** 2024-01-20

**Authors:** Jianzhi Huang, Yueyun Sheng, Pengfei Xue, Donghui Yu, Peng Guan, Jiangang Ren, Wenxi Qian

**Affiliations:** 1College of Animal Science and Technology, Tarim University, Alar 843300, China; huangjianzhi2021@126.com (J.H.); 15999220945@163.com (Y.S.); 18189388984@163.com (P.X.); 18153443316@163.com (D.Y.); guanpengmhl@163.com (P.G.); renjiangang0503@163.com (J.R.); 2Key Laboratory of Tarim Animal Husbandry Science & Technology, Xinjiang Production & Construction Group, Alar 843300, China

**Keywords:** Tarim wapiti, rumen sac, microorganisms, histomorphology, fermentation parameters

## Abstract

The rumen is divided into multiple rumen sacs based on anatomical structure, and each has its unique physiological environment. Tarim wapiti preserved roughage tolerance after domestication, and adaptation to the desertified environment led to the development of a unique rumen shape and intraruminal environment. In this work, six Tarim wapiti were chosen and tested for fermentation parameters, microbes, and histomorphology in four rumen areas (Dorsal sac, DS; Ventral sac, VS; Caudodorsal blind sac, CDBS; Caudoventral blind sac, CVBS). Tarim wapiti’s rumen blind sac had better developed rumen histomorphology, the ventral sac was richer in VFAs, and the dominant bacteria varied most notably in the phylum *Firmicutes*, which was enriched in the caudoventral blind sac. The ventral sac biomarkers focused on carbohydrate fermentation-associated bacteria, the dorsal sac focused on N recycling, and the caudoventral blind sac identified the only phylum-level bacterium, *Firmicutes*; we were surprised to find a probiotic bacterium, *Bacillus clausii*, identified as a biomarker in the ventral sac. This research provides a better understanding of rumen fermentation parameters, microorganisms, and histomorphology in the Tarim wapiti rumen within a unique ecological habitat, laying the groundwork for future regulation targeting the rumen microbiota and subsequent animal production improvement.

## 1. Introduction

The Tarim wapiti is primarily found around the Tarim River and its tributaries, which are in the arid Tarim Basin of the Xinjiang Uygur Autonomous Region in China. This subspecies of Chinese deer is unique in that it is adapted to living in desertified habitats [1]. However, the Tarim wapiti has developed a strong adaptability to forage with poor quality, low protein content, and high crude fiber content due to long-term natural selection pressure under ecological adversity [2,3]. Qian et al. [3] reported that the Tarim wapiti showed significant advantages in nutritional digestibility and rumen fermentation above other ruminants (cattle and sheep), as well as unique dominant bacteria for *Oscillospira* [4]. It is possible that the Tarim wapiti’s capacity to adapt to its unique natural ecological context resulted in the maintenance of a distinct rumen histomorphology and intraruminal environment even after domestication.

Ruminants rely on the diverse microbial communities that inhabit the rumen to break down non-directly accessible fiber into small molecules that the organism can use. The rumen is divided into regions, each with a unique region of microbial and histological features. Bacteria account for around 50% to 80% of all microorganisms in the rumen [5]. Bacterial communities are characterized as fiber-adherent bacterial communities, epimural bacteria, or planktonic bacterial communities based on their biological niche in the rumen [6]. The liquid-phase microorganisms accounted for around 30% of total rumen bacteria, with *Prevotella* and *Bacteroides* being the most abundant genera [7]. The fermentation of soluble carbohydrates yields the generation of VFAs (volatile fatty acid, VFAs) [8], MCP (microbial proteins, MCP) [9], and other minute molecules.

Furthermore, the developmental consequences of rumen morphology and structure affect nutrient digestion and absorption. The size, amount, and distribution of rumen papillae in the rumen are all closely related to feeding, digestibility, and energy metabolism. [10]. Simultaneously, the rumen epithelium is a distinct zone of host–microorganism interaction that is critical to host nutrition, energy metabolism, and immune response [11]. The variety of rumen liquid-phase microorganisms [12], pH [13], NH_3_-N, VFAs [14], and rumen histomorphology [15], varies amongst rumen regions. It has also been shown that the variety of microorganisms in rumen chyme is independent from the region of the sample [16]. Inconsistent results from microorganisms, fermentation parameters, and histomorphological structures at various regions of the rumen might be attributed to the animal’s diet dictating the condition and distribution of rumen contents. Red deer are mixed feeders based on their diet and should have a specific rumen structure and intraruminal environment [17]. Furthermore, most studies on the diversity of various regions in Tarim wapiti rumen have focused on histomorphology [15,18,19], with just a few studies on microbes and fermentation characteristics.

Given the previous single region of rumen sampling in deer species, there is a scarcity of research on the complete rumen structure. Consequently, this research investigated rumen microorganisms, fermentation parameters, and histomorphology in the rumens of six Tarim wapiti from four various regions. On the one hand, we learned about the structure and intraruminal environment of the Tarim wapiti’s rumen in its unique ecological environment; on the other hand, by sampling the rumen at various regions, we laid the groundwork for future research on the interactions between the Tarim wapiti’s microorganisms and its host.

## 2. Materials and Methods

### 2.1. Animal and Feeding Management

The Tarim wapiti used in this study were selected from the 31st regiment field of the Second Division of Xinjiang Production and Construction Corps in China, and the experiment was conducted on 20 December 2022. The study included three healthy adult males (200 ± 5.0 kg, 5 years old) and three healthy adult females (175 ± 5.0 kg, 5 years old). The study area, located at geographical coordinates 86°45′–87°00′ E, 40°49′–40°59′ N, has an altitude ranging from 820 to 1100 m above sea level and experiences a typical continental arid climate. The deer were provided with two feedings per day at 9 am and 7 pm and had ad libitum access to feed and water. The basal diets consisted of a 3:7 ratio of concentrate to roughage, and the composition and nutrient levels of these diets can be found in Table 1. The concentration of CP, Ca, and P in the feed supplied was determined via reference to the AOAC method [20]. CP was determined by Kjeldahl nitrogen determination, Ca was determined by potassium permanganate titration, and P was determined by absorbance photometry. It is worth noting that the enclosure was diligently maintained, with proper ventilation, cleanliness, and regular disinfection measures in place.

### 2.2. Collection of Samples

The deer were slaughtered in the morning after two hours of feeding, with all deer retained left side up and the rumen isolated from the other organs in the abdominal cavity; the rumen was set somewhat flat onto a clean tray, and sterile cotton thread was used to isolate the region by ligation to prevent the rumen from shaking and mixing the contents. After isolating the rumen, each region was cut from the middle position with sterilized surgical scissors, and rumen fluid samples were collected from four regions of the rumen, including the dorsal sac (DS), ventral sac (VS), cephalic dorsal blind sac (CDBS), and caudal ventral blind sac (CVBS), using hands with sterile gloves [21].

The rumen contents were filtered through four layers of sterile gauze, and the pH was immediately measured before being stored in a 5 mL strain preservation tube at -80 degrees Celsius to be analyzed for 16S rDNA and rumen fermentation parameters of rumen liquid. In accordance with the anatomical structure of the rumen [21], tissue blocks of 2 × 2 cm^2^ in the middle were obtained from various regions. Rumen tissue was cleaned with 0.9% physiological salt water and fixed in a 4% paraformaldehyde solution before being assessed for rumen papilla density and then sectioned. The rumen liquid was sent to Beijing Novogene Bioinformatics Technology Co., Ltd. (Beijing, China) for sequencing analysis of the 16S rRNA V3–V4.

### 2.3. Sectioning and Analyzing Ruminal Tissue

A measurement of 1 × 1 cm^2^ of rumen tissue was taken from various regions, and the density of rumen papillae was examined with a dissection microscope (SDPTOP SZ, SUNNY HENGPINGINSTRUMENT, Shenzhen, China).

The rumen tissue was flushed (24 h), dehydrated (50% alcohol for 60 min; 60% alcohol for 60 min; 75% alcohol for 60 min; 85% alcohol for 60 min; 95% alcohol for 30 min; 100% alcohol for 30 min), and immersed in wax (58 °C paraffin wax for 1 h; 60 °C paraffin for 1 h); embedding (HD-3108, huidayiqi, Xiaogan, Hubei, China) and cutting 7 μm continuous sections using a semi-automatically rotary slicer (RM2016, Leica Microsystems(Shanghai), Shanghai, China) was then undertaken, and the sections were unfolded, dried for 2 h at 60 °C, stained with HE, sealed, and inspected using an electron microscope. ImageJ (Version 1.410) was used to formulate the measurements. For each rumen sample, three randomly selected discontinuous regions were sectioned, and five discontinuous and random locations were selected for rumen papilla height, papilla breadth, and muscle layer thickness measurements.

*SEF* (Surface enlargement factor of rumen mucosa) was calculated by the Hofmann [18] method.
$$SEF=\left(RO+100\right)/100$$

ZZ: rumen papillae count (cm^2^); ZL: rumen papillae height (mm); ZB: rumen papillae breadth (mm); ZO: average rumen papillae surface area (mm^2^) = ZL × ZB × 2; ZO: average rumen papillae surface area (mm^2^) = ZL × ZB × 2.

### 2.4. Analysis of Rumen Fermentation Parameters

The pH of rumen liquid was measured using a portable pH meter (FE28, Mettler Toledo, Shanghai, China). The NH_3_-N content was measured using Broderick’s alkaline sodium hypochlorite–phenol spectrophotometry [22]. The quantification of VFAs (acetate, propionate, butyrate, and valerate) in each fraction was performed using gas chromatography (SP7800, Beijing Jingke Ruida, Beijing, China). The parameters for the gas chromatography analysis were set as follows: column temperature = 120 °C, injector temperature = 230 °C, detector temperature = 250 °C, and injection volume = 2 μL.

### 2.5. Rumen Liquid 16S rRNA Gene Analysis

#### 2.5.1. DNA Extraction and PCR Products, Acquisition, Quantification, and Qualification

The genomic DNA was extracted using a DNA extraction kit (TianGen, Beijing, China), and the purity and concentration of the DNA were detected by 1% agarose gel electrophoresis. The V3-V4 region of 16S rDNA was amplified using the universal primers 341F (GTGCCAGCMGCCGCGGTAA) and 806R (GGACTACHVGGGTWTCTAAT). All PCR reactions were carried out with 15 µL of Phusion^®^ High-Fidelity PCR Master Mix (New England Biolabs, Ipswich, MA, USA), 0.2 µM of forward and reverse primers, and approximately 10 ng of template DNA. Thermal cycling consisted of initial denaturation at 98 °C for 1 min, followed by 30 cycles of denaturation at 98 °C for 10 s, annealing at 50 °C for 30 s, and elongation at 72 °C for 30 s, followed by a final elongation step at 72 °C for 5 min. It is important to note that, when genomic nucleic acids are amplified, each nucleic acid is amplified as a separate sample with a separate barcode to ensure that the amplified products are distinguishable when mixed, and therefore no distinction is made between duplicate parallel samples when mixing the samples for library construction. The same volume of 1× loading buffer (contained SYB green) was mixed with PCR products and electrophoresis was operated on 2% agarose gel for amplicon detection. PCR products were mixed in equimolar amounts. Then, the mixture of the PCR products was purified with a Universal DNA Purification Kit (TianGen, China, Catalog #: DP214).

#### 2.5.2. Library Preparation and Sequencing

Sequencing libraries were generated using NEB Next^®^ Ultra™ II FS DNA PCR-free Library Prep Kit (New England Biolabs, USA, Catalog #: E7430L) following the manufacturer’s recommendations, and indexes were added. The library was checked with Qubit and real-time PCR for quantification and bioanalyzer for size distribution detection. Quantified libraries were pooled and sequenced on a NovaSeq 6000 instrument, according to the effective library concentration and data amount required.

#### 2.5.3. Paired-End Reads Assembly and Quality Control

Paired-end reads were merged with FLASH (V1.2.11, http://ccb.jhu.edu/software/FLASH/, accedded on 12 April 2023) to produce Raw Tags [23]. Quality filtering on the raw tags was performed using the fastp (Version 0.23.1) software to obtain high-quality Clean Tags [24]. The Clean Tags were compared with the reference database (Silva database (16S), https://www.arb-silva.de/, accessed on 12 April 2023), using UCHIME Algorithm for the detection and removal of chimeric sequences to ensure that effective Tags were obtained [25].

#### 2.5.4. ASVs Denoise and Taxonomic Annotation

For the Effective Tags obtained previously, denoise was performed with DADA2 in the QIIME2 software (Version QIIME2-202006) to obtain initial ASVs (Amplicon Sequence Variants), and then ASVs with abundance less than 5 were filtered out [26]. Taxonomic annotation was performed using QIIME2 software. For the 16S rRNA gene, the annotation database was Silva Database. LEfSe (LDA Score = 4) was used to analyze overall differences in microbial composition.

#### 2.5.5. Alpha Diversity and Beta Diversity

Alpha diversity indices of the samples were calculated, including observed ASVs, Shannon and Chao1, using the QIIME2 software. R (version 3.5.3) with the ggplot2 package and the ade4 package for PCoA based on Bray–Curtis distance were used.

### 2.6. Statistical Analysis

Data on rumen fermentation parameters, histomorphology, microbial diversity, and variation in relative abundance at the phylum, family, and genus levels were analyzed with one-way ANOVA and multiple comparisons through Duncan’s method, with *p* < 0.05 indicating significant differences.

## 3. Results

### 3.1. Analysis of Histomorphological Variations in Rumen Regions

The rumen histomorphology varied substantially between regions (Figure 1). The CVBS and CDBS papillae were blade-shaped, whereas the DS and VS papillae were cone-shaped (Figure 2). The DS papillae density (63.33/cm^2^) was significantly higher than that of the CDBS and CVBS (*p* < 0.05). The CDBS had significantly more height, breadth, and *SEF* of papillae than the other regions (*p* < 0.05), and the CVBS and CDBS had higher *SEF* than the DS. The DS had the thickest muscle layer (3.61 mm), which was significantly higher than the other regions and varied significantly (*p* < 0.05) among all four regions. (Table 2).

### 3.2. Analysis of Variations in Fermentation Characteristics at Several Segments in the Rumen

Differences in pH, NH_3_-N, acetate, propionate, butyrate, valerate, and TVFA in various regions of the rumen of the Tarim wapiti were not significant (*p* > 0.05), with the lowest levels in the VS of pH (5.92) and NH_3_-N (12.53 mg/100 mL); VFAs were highest in the VS and lowest in the CDBS (Table 3).

### 3.3. Analysis of Bacterial Communities Based on the 16S rRNA Sequencing

#### 3.3.1. Analysis of Effective Sequences, Alpha Diversity, and Beta Diversity of Bacteria in Several Rumen Regions

There were 1,862,526 Effective Tags obtained, with an average of 77,605.25 per sample, and 39,623 ASVs created by clustering them with 100% similarity. The differences in the observed ASVs, Chao1 index, and Shannon index in various regions of the rumen were not statistically significant but were highest in the DS and lowest in the VS (Table 4). Using PCoA analysis based on the Bray–Curtis distance, we observed that there was no significant variation between the liquid microbial populations in various regions of the rumen. (Figure 3).

#### 3.3.2. The Composition of Bacterial Taxa in Various Regions of the Rumen

In this study, 41 phyla, 360 families, and 718 genera were identified. *Bacteroidota* and *Firmicutes* were the dominant phyla at various regions in the rumen of Tarim wapiti, which accounted for more than 80% of the different phyla in the rumen. *Bacteroidota* showed the highest proportion in the CDBS (51.53%) and the lowest in the VS (42.39%). *Firmicutes* relative abundance was significantly higher in the DS (40.36%) and CVBS (44.37%) than in the CDBS (29.31%) (*p* < 0.05). The relative abundance of *Proteobacteria* in numerous rumen areas surpassed 5%, whereas their abundance in VS and CDBS exceeded 13%. Furthermore, the relative abundance of *Actinobacteriota* was significantly higher in the DS than in the VS and CVBS (*p* < 0.05) (Table 5).

*Rikenellaceae*, *Prevotellaceae*, and *Lachnospiraceae* were the most prevalent three bacterial families in terms of relative abundance at various rumen regions, with relative abundances of more than 10% for *Rikenellaceae* and *Prevotellaceae* and less than 5% for *Lachnospiraceae*. Furthermore, the CDBS had a higher proportion of *Xanthomonadaceae* (8.15%) (Table 6). 

At the genus level, the dominant genera in the rumen of Tarim wapiti were the *Rikenellaceae RC9 gut group* and *Prevotella*, with the relative abundance of *Rikenellaceae RC9 gut group* exceeding 10%, and the CDBS had a relatively high abundance of *Stenotrophomonas*, accounting for 8.13% (Table 7). 

#### 3.3.3. LEfSe Analysis of Bacterial Communities at Various Rumen Regions

The number of relevant biomarkers with significant variations in VS, DS, CVBS, and CDBS at different levels of taxonomic units was 9, 8, 2, and 1, respectively (Figure 4). The VS contained biomarkers from the *RF39* order, *RF39* family, *RF39* genus, *NK4A214 group* genus, *Bacillus clausii* from *Firmicutes*, *Acetobacterales*, *Acetobacteraceae*, *Acetobacter* from *Proteobacteria*, and *Desulfovibrio* from *Desulfobacterota*. *Nitrosomonadaceae*, *Rhodocyclaceae*, *Moraxellaceae*, and *Acinetobacter* biomarkers were found in the DS, as well as *MND1* from *Proteobacteria*, *Actinobacteria*, *Micrococcales*, and *Micrococcaceae* from *Actinobacteriota*. *Firmicutes* and *Negativicutes* were discovered as biomarkers in the CVBS. A biomarker discovered in the CDBS is *Aeromicrobium* from *Actinobacteriota* (Table 8).

## 4. Discussion

### 4.1. Histomorphological Variations in Rumen Regions

The architecture of the ruminant digestive system evolved as result of adaptation to the natural ecological environment. The architecture and structure of the rumen influence nutrient digestion and absorption. Rumen papillae size, number, and position in the rumen have been linked to animal foraging, digestibility, and energy metabolism [10]. Animal anatomy can be modified by species and the type of forage consumed [27]. The height and breadth of the papillae in the Cervus elaphus cranial sac and blind sac were discovered to be higher than those in the dorsal sac [19]. Mason et al. [15] discovered no differences in rumen papilla density, height, and breadth between dorsal and ventral sacs in cultivated Cervus elaphus and that cranial pillar papillae had higher and broader heights and breadths than the dorsal and ventral sacs, but papilla densities and rumen development indicators were lower in farmed Cervus elaphus than in in wild Cervus elaphus [15,19]. Furthermore Hofmann et al. [18] reported that the papillae in the rumen of wild Cervus elaphus varied by region, with rumen papillae densities varying from 38.6 to 48.0 papillae/cm^2^, papillae heights ranging from 2.8 to 10.27 mm, and papillae breadths ranging from 0.75 to 1.84 mm. The morphology and density of papillae in various regions of Tarim wapiti rumen were consistent with previous studies, and it is noteworthy that we found that the height and breadth of Tarim wapiti rumen papillae were not only more than those of farmed Cervus elaphus but also more than those of wild Cervus elaphus [15,19]. This finding implies that the persistence of higher papillae and broader papillae structure in the Tarim wapiti rumen may be one of the main explanations for the animals’ high adaptability to their unique ecological environment (desertification, high salinity) and the preservation of roughage tolerance characteristics after domestication.

To complete the rumination procedure, several regions in the rumen (dorsal sac, ventral sac, caudodorsal blind sac, and caudoventral blind sac) and the reticulum are interrelated. Secondary rumen contractions include just a portion of the rumen, such as the dorsal coracobrachialis column, the dorsal sac, and the caudodorsal blind sac, and are often restricted to the dorsal sac solely [28]. Rumen volume, weight expansion, and peristalsis might all be improved by increasing the thickness of the rumen muscle layer. In this study, the dorsal sac’s muscular layer was the thickest, while the caudodorsal blind sac was the thinnest, indicating that secondary contractions in the Tarim wapiti’s non-ruminant phase are largely driven by the dorsal sac for rumen peristalsis. Furthermore, Hofmann’s theory tackles the density of papilla distribution and the surface expansion factor of rumen mueoa in various regions of rough feeder rumen [29]. Prins and Geelen [30] studied the *SEF* of red deer, fallow deer, and roe deer and discovered that the *SEF* of rough feeders was 3.5, and that of concentration feeders was in the range of 7.0–9.5. Mason et al. [15] observed that the *SEF* of red deer ranged from 4.99 to 14.73. Mason’s research discovered that the *SEF* of horse deer ranged from 4.99 to 14.73. The overall *SEF* of the Tarim wapiti in the study was 4.76 ± 1.71, which was consistent with the *SEF* of 3.5–7.0 in the mixed feeders, and there were also significant differences in *SEF* and papilla densities in various regions of the rumen, which was also consistent with Hofmann’s hypothesis that the *SEF* was significantly higher in the caudodorsal blind sac than in the other regions, while papillary densities in the dorsal and ventral sacs were much higher than in the caudodorsal blind sac.

### 4.2. Variations in Fermentation Characteristics at Several Regions in the Rumen

Rumen pH is the result of synergistic acid–base regulation by the rumen microbial communities and host rumen metabolism [31]. Rumen pH and VFA content were discovered to be adversely linked [32]. In this study, pH and acetate, propionate, butyrate, valerate, and TVFA showed negative correlation linkages in the four regions of Tarim wapiti’s rumen, which corresponded to the pattern of change. Only *Firmicutes* differed among the microorganisms producing VFAs by fermentation in various regions of the Tarim wapiti’s rumen, and the morphology and density of the papillae differed considerably, but the content of VFAs and pH did not. The size and quantity of rumen papillae, as well as their location in the rumen, were directly associated with energy metabolism. Consequently, the metabolic level of rumen papillary epithelial cells in Tarim reindeer may be a significant determinant affecting rumen VFAs content and pH.

### 4.3. The Composition of Bacterial Communities in Various Rumen Regions

Microbial communities in the rumen’s various ecological environments change due to interactions with the host and changes in production, and the rumen liquid-phase microbial community has regular access to various types of feeds due to constant redistribution within the rumen [33]. *Bacteroides* was found to be more dominant in the liquid phase of the rumen contents of Cervus canadensis and white-tailed deer, whereas *Firmicutes* was the most abundant phylum in the solid phase, and *Prevotella* was the co-dominant genus in both the solid and liquid phases of the rumen contents, but its relative abundance was higher in the liquid phase than the solid phase [34]. *Firmicutes* are abundant in both the solid and liquid phase of roe deer Capreolus rumen [35]. In this current study, we discovered that the dominant bacteria in the liquid phase of Tarim wapiti rumen contents were the same as in previous microbiological studies of deer rumen contents, and that, with a 3:7 ratio of concentrate to roughage in the current diet, the dominant bacteria (*Prevotellaceae*, *RikenellaceaeRC9 gut group*, *Prevotella*) were the same as those in Rangifer tarandus [36], and the major bacterial species in the rumen contents of Tarim wapiti [2] fed a high-fiber feed (concentrate/crude ratio of 2:8) were consistent. This demonstrates that rumen bacteria in the Tarim wapiti remain rather stable throughout roughage adaptation, preserving the efficiency of energy absorption from plant fibers.

Ruminants are classed as concentrate selectors, grass and roughage eaters, or intermediate, opportunistic, mixed feeders based on the type of plant they intake and the anatomy of their digestive system [27]. Among such are the contents of the rumen digestive metabolism of concentrate selectors in their natural state, with no obvious solid, liquid, or gas stratification occurrences [37]. The Bovidae family eats grass and roughage, and its rumen contents are visibly stratified [17]. The microbial communities in bovine rumen chow did not differ by sampling region [16], but there were variations in the structure and composition of microorganisms adhering to the rumen epithelium (*Firmicutes*, *Bacteroides*, *Prevotellaceae*, *Lachnospiraceae*) [38] as well as significant differences in the microbial composition of the rumen contents in both the solid phase and the liquid phase [39]. The red deer is a mixed feeder that is intermediate, opportunistic, and rotates between concentrate pickers and grass and roughage eaters [17]. The liquid and solid phase microbial populations in Cervus canadensis rumen contents were not taxonomically categorized [34]. The dominant bacteria in Tarim wapiti rumen fluid from various regions were consistent with the dominant bacteria in Tarim wapiti rumen fluid previously collected by oral intubation [2,4], and they were consistent with the dominant bacteria in winter Rangifer tarandus rumen contents collected by slaughtering methods [36]. Furthermore, only *Firmicutes* differed in relative abundance in various regions among the dominant rumen liquid phase microorganisms in the Tarim wapiti, suggesting that the Tarim wapiti rumen contents were also unobtrusively stratified, and that *Firmicutes* is one of the major bacteria affecting rumen nutrient digestion.

Qian et al. [2] reported that the rumen fluid of Tarim wapiti fed low-fiber diets (concentrate/crude ratios of 4:6, 5:5) contained a greater abundance of the *RF* bacteria family, and that the number increased as dietary fiber increased. All members of the *Bacilli* genus may produce enzymes to break down polysaccharides and ferments to produce organic acids such as acetic acid and lactic acid. The *NK4A214 group* and *Christensenellaceae R-7 group* account for around half of total hydrogenase transcript expression in the rumen and are the primary electron sources during methane production [40]. *Desulfovibrio* is a sulfate-reducing bacteria that produces hydrogen sulfide by degrading chemicals such as VFAs and amino acids [41]. The *RF39* order, *RF39* family, *RF39* genus, *NK4A214 group*, and *Desulfovibrio* were identified as biomarkers of the Tarim wapiti’s rumen ventral sac in this study, and these microorganisms may voluntarily play an important role in the degradation of soluble carbohydrates and the production of greenhouse gases. Furthermore, a single species-level biomarker “*Bacillus clausii*” was discovered in the rumen ventral sac, which is a member of a group of acid- and saline-resistant probiotic bacteria [42]. Probiotics have been demonstrated to stimulate epithelial cells, enhance mucus production, restore mucus layer thickness, and reduce permeability [43].

It was discovered that *Rhodocyclaceae* has nitrate reduction, aromatic chemical degradation, and a good nitrogen fixation function [44]. *Nitrosomonadaceae* mostly oxidize ammonia and facilitate nitrogen cycle [45]. In this work, *Rhodocyclaceae* and *Nitrosomonadaceae* were discovered as rumen dorsal sac biomarkers, showing that rumen dorsal sac liquid phase bacteria play a role in nitrogen use throughout the rumen. *Firmicutes* have been found to be effective fiber degraders [46]. Primary contractions in the rumen of ruminants follow the dorsal sac, caudodorsal blind sac, caudoventral blind sac, ventral sac, and cranial sac at the end of a cycle [33]. *Firmicutes* and *Negativicutes* were identified as rumen caudoventral blind sac biomarkers in this study, indicating that the liquid phase microorganisms of the rumen caudoventral blind sac provide more digestible fiber and VFAs to the rumen ventral sac, which is the main region of rumen absorption. *Aeromicrobium*, an antibiotic-producing probiotic, was identified as a biomarker of the caudodorsal blind sac in this study. Other probiotic organisms were discovered as biomarkers in various parts of the rumen.

## 5. Conclusions

Tarim wapiti’s rumen blind sac had better developed rumen histomorphology, the ventral sac was richer in VFAs, and the dominant bacteria varied most notably in the phylum *Firmicutes*, which was enriched in the caudoventral blind sac. The ventral sac biomarkers focused on carbohydrate fermentation-associated bacteria, the dorsal sac focused on N recycling, and the caudoventral blind sac identified the only phylum-level bacterium, *Firmicutes*, and we were surprised to find a probiotic bacterium, *Bacillus clausii*, identified as a biomarker in the ventral sac.

These findings contribute to a better knowledge of Tarim wapiti rumen anatomy and an intraruminal environment in a desertified ecology. This will allow for the future sampling of various regions in the Tarim wapiti rumen to further investigate microbial–host interactions within the rumen.

## Figures and Tables

**Figure 1 microorganisms-12-00216-f001:**
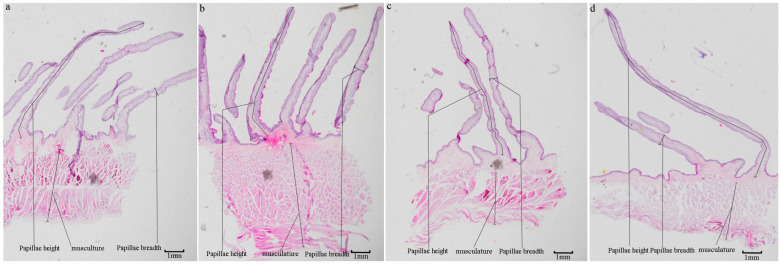
Rumen slices at various regions (20×). (**a**): VS; (**b**): DS; (**c**): CVBS; (**d**): CDBS.

**Figure 2 microorganisms-12-00216-f002:**
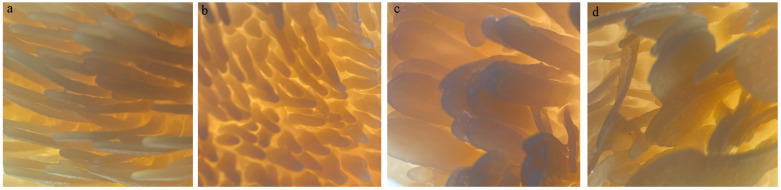
Rumen papilla density at various regions (10×). (**a**): VS; (**b**): DS; (**c**): CVBS; (**d**): CDBS.

**Figure 3 microorganisms-12-00216-f003:**
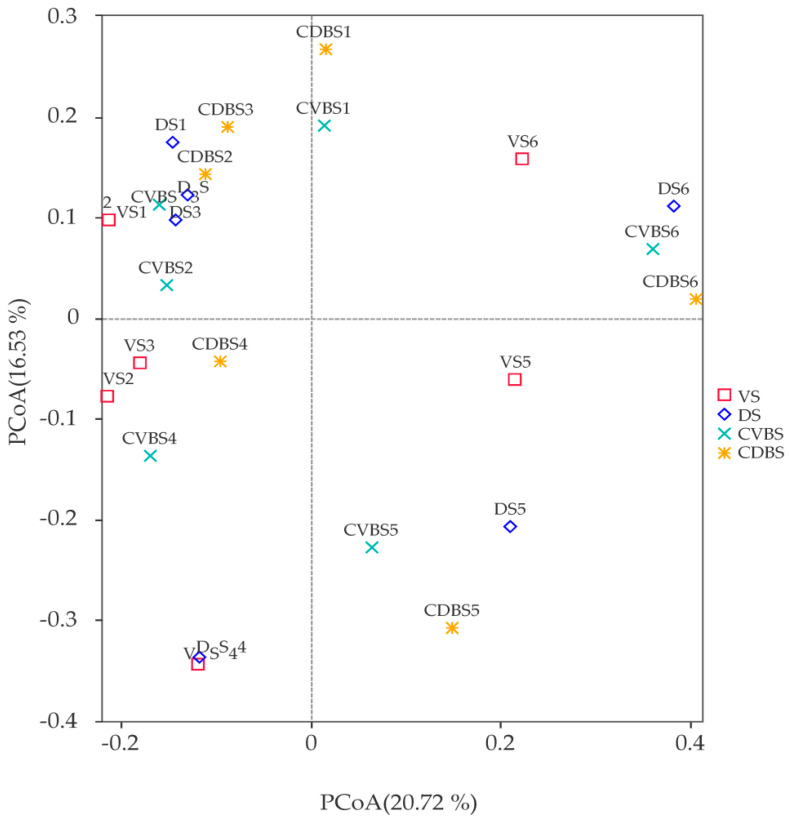
A 2D PCoA plot of microbial communities in the rumen at several regions (*n* = 6 VS; *n* = 6 DS; *n* = 6 CVBS; *n* = 6 CDBS); 1–3 for females and 4–6 for males.

**Figure 4 microorganisms-12-00216-f004:**
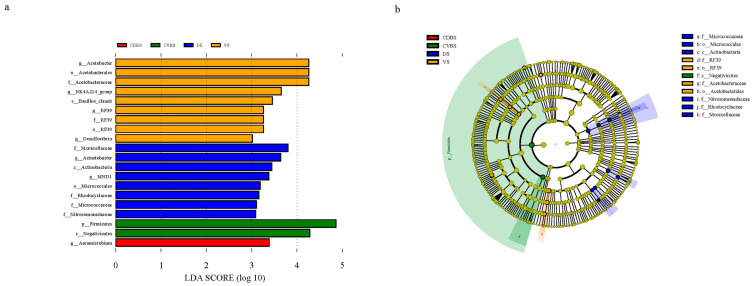
(**a**) LDA value > 3. (**b**) The classification level of the circle from the outside to the inside is in order of phyla, class, order, family, genus, and species. Different color points in the phylogenetic tree stand for the bacteria which are important in each group, respectively. Yellow points indicate taxa, which have no significant difference among groups.

**Table 1 microorganisms-12-00216-t001:** Composition and nutritional level of diets.

Items	Value
Composition	Content (%)
forage	
Corn stalks	19.5
Wheat stalks	13
Alfalfa	22.5
Cottonseed hull	15
concentrate	
Corn	23.5
Soybean meal	2.5
Cotton meal	2.5
Salt	0.5
Premix ^¶^	1
Total	100
Nutrient Levels	
Digestible energy (DE) ^§^, MJ/kg	11.1
Crude protein (CP), %	11.51
Calcium (Ca), %	0.53
Phosphorus (P), %	0.24

^¶^ Premix: VA 940 IU/kg, VE 20 IU/kg, minerals (mg/kg), S 200 mg/kg, Fe 24 mg/kg, Cu 8 mg/kg, Mn 40 mg/kg, Zn 40 mg/kg, I 0.3 mg/kg, Se 0.2 mg/kg, Co 0.1 mg/kg. ^§^ DE: DE was calculated value.

**Table 2 microorganisms-12-00216-t002:** Variance in histomorphology between rumen regions (*n* = 6 VS; *n* = 6 DS; *n* = 6 CVBS; *n* = 6 CDBS).

Items	Rumen Regions				SEM	*p*-Value
	VS	DS	CVBS	CDBS		
Papilla density number/cm^2^	60.33 ^ab^	63.33 ^a^	55.33 ^bc^	52.83 ^c^	1.88	0.002
Papillae height/mm	5.77 ^bc^	4.90 ^c^	6.60 ^b^	9.03 ^a^	0.04	<0.01
Papillae breadth/mm	0.45 ^b^	0.37 ^c^	0.48 ^b^	0.65 ^a^	0.02	<0.01
Muscle layer/mm	2.52 ^b^	3.61 ^a^	1.71 ^c^	1.43 ^d^	0.02	<0.01
*SEF*	4.10 ^bc^	3.24 ^c^	4.51 ^b^	7.20 ^a^	0.34	<0.01

Different letters indicate significant differences (*p* < 0.05).

**Table 3 microorganisms-12-00216-t003:** The fermentation parameters at several regions in the rumen (*n* = 6 VS; *n* = 6 DS; *n* = 6 CVBS; *n* = 6 CDBS).

Items	Rumen Regions				SEM	*p*-Value
	VS	DS	CVBS	CDBS		
pH	5.92	5.96	5.95	5.97	0.01	0.289
NH_3_-N mg/100 mL	12.53	12.57	12.57	12.56	0.11	0.999
Acetate mmol/L	59.15	58.28	57.64	57.13	0.59	0.674
Propionate mmol/L	11.78	11.67	11.33	11.32	0.14	0.954
Butyrate mmol/L	3.09	2.80	2.76	2.68	0.1	0.428
Valerate mmol/L	1.51	1.34	1.29	1.15	0.19	0.936
TVFA	75.53	74.09	73.01	72.28	0.5	0.212

**Table 4 microorganisms-12-00216-t004:** Alpha diversity indices for several rumen regions (*n* = 6 VS; *n* = 6 DS; *n* = 6 CVBS; *n* = 6 CDBS).

Items	Rumen Regions				SEM	*p*-Value
	VS	DS	CVBS	CDBS		
Observed ASVs	1525.00	1749.17	1673.67	1656.00	42.52	0.32
Chao1 index	1530.85	1747.66	1677.36	1653.65	42.11	0.34
Shannon index	8.81	9.27	9.12	8.74	0.15	0.57

**Table 5 microorganisms-12-00216-t005:** The relative abundance of microbial phyla in the various rumen regions (*n* = 6 VS; *n* = 6 DS; *n* = 6 CVBS; *n* = 6 CDBS).

Phyla	Rumen Regions				SEM	*p*-Value
	VS	DS	CVBS	CDBS		
Bacteroidota	42.39	44.86	44.34	51.53	2.47	0.613
Firmicutes	37.82 ^ab^	40.36 ^a^	44.37 ^a^	29.31 ^b^	1.9	0.024
Proteobacteria	13.26	8.37	7.27	13.22	2.08	0.658
Spirochaetota	0.31	0.63	0.19	1.03	0.23	0.608
Patescibacteria	1.73	1.40	0.86	1.02	0.19	0.378
Synergistota	0.86	0.42	0.34	0.56	0.14	0.594
Cyanobacteria	0.95	0.75	0.28	0.35	0.14	0.263
Euryarchaeota	0.73	0.50	0.43	0.48	0.08	0.583
Desulfobacterota	0.83	0.51	0.86	0.87	0.07	0.222
Actinobacteriota	0.49 ^b^	1.02 ^a^	0.39 ^b^	0.67 ^ab^	0.08	0.021

Different letters indicate significant differences (*p* < 0.05).

**Table 6 microorganisms-12-00216-t006:** The relative abundance of microbial families in the various rumen regions (*n* = 6 VS; *n* = 6 DS; *n* = 6 CVBS; *n* = 6 CDBS).

Families	Rumen Regions				SEM	*p*-Value
	VS	DS	CVBS	CDBS		
Rikenellaceae	12.27	14.15	13.89	16.57	1.25	0.905
Prevotellaceae	15.93	13.56	15.64	15.46	0.75	0.697
Lachnospiraceae	6.82	6.67	8.28	6.11	0.48	0.452
F082	3.48	4.28	3.66	4.61	0.23	0.371
Oscillospiraceae	4.41	4.89	4.81	3.87	0.20	0.268
Ruminococcaceae	4.55	5.31	4.24	2.32	0.65	0.442
Xanthomonadaceae	5.57	1.88	3.01	8.15	1.77	0.629
p-251-o5	1.95	3.67	2.83	4.18	0.81	0.8
Acetobacteraceae	3.79	0.26	1.66	0.05	0.85	0.407
Bacteroidales BS11 gut group	2.32	1.04	1.80	1.83	0.54	0.887

**Table 7 microorganisms-12-00216-t007:** The relative abundance of microbial genera levels in the rumen at various regions (*n* = 6 VS; *n* = 6 DS; *n* = 6 CVBS; *n* = 6 CDBS).

Genera	Rumen Regions				SEM	*p*-Value
	VS	DS	CVBS	CDBS		
Stenotrophomonas	5.57	1.86	3.01	8.13	1.77	0.629
Rikenellaceae RC9 gut group	12.11	14.02	13.61	16.41	1.25	0.704
Acetobacter	3.79	0.21	1.65	0.03	0.86	0.4
Ruminococcus	1.94	3.67	2.84	4.18	0.63	0.471
p251o5	1.94	3.67	2.84	4.18	0.81	0.8
Prevotella	9.16	6.92	8.53	8.35	0.57	0.581
Bacteroidales BS11 gut group	2.32	1.04	1.80	1.83	0.54	0.887
Quinella	0.98	3.07	3.84	1.10	0.46	0.055
Muribaculaceae	3.26	3.89	3.55	4.25	0.35	0.792
Eubacterium coprostanoligenes group	3.33	2.38	3.10	1.52	0.30	0.137

**Table 8 microorganisms-12-00216-t008:** Biomarkers and relative abundance of several rumen regions (*n* = 6 VS; *n* = 6 DS; *n* = 6 CVBS; *n* = 6 CDBS).

Rumen Regions/Taxa	Phylum %	Class %	Order %	Family %	Genus %	Species %
VS	Firmicutes	Bacilli	RF39 0.61%	RF39 0.61%	RF39 † 0.61%	—
	Proteobacteria	Alphaproteobacteria	Acetobacterales 3.79%	Acetobacteraceae 3.79%	Acetobacter † 3.79%	—
	Desulfobacterota	Desulfovibrionia	Desulfovibrionales	Desulfovibrionaceae	Desulfovibrio † 0.39%	—
	Firmicutes	Clostridia	Oscillospirales	Oscillospiraceae	NK4A214 group † 2.26%	—
	Firmicutes	Bacilli	Bacillales	Bacillaceae	Bacillus	Bacillus clausii † 0.60%
DS	Actinobacteriota	Actinobacteria 0.69%	Micrococcales 0.31%	Micrococcaceae † 0.20%	—	—
	Proteobacteria	Alphaproteobacteria	Rhodobacterales	Rhodobacteraceae † 0.35%	—	—
	Proteobacteria	Gammaproteobacteria	Pseudomonadales	Moraxellaceae 1.08%	Acinetobacter † 0.69%	—
	Proteobacteria	Gammaproteobacteria	Burkholderiales	Nitrosomonadaceae 0.10%	MND1 † 0.04%	—
CVBS	Firmicutes 44.40%	Negativicutes † 7.18%	—	—	—	—
CDBS	Actinobacteria	Actinobacteria	Propionibacteriales	Nocardioidaceae	Aeromicrobium † 0.0026%	—

Biomarkers are those with relative abundance share values. † represents a key biomarker.

## Data Availability

The data that support the findings of this study are openly available in NCBI. http://www.ncbi.nlm.nih.gov/bioproject/1048862, accessed on 31 December 2023.

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
