# Peer review of "Patterns of Spatial Variation in Rumen Microbiology, Histomorphology, and Fermentation Parameters in Tarim wapiti (Cervus elaphus yarkandensis)"

_microorganisms, 2024, doi:10.3390/microorganisms12010216_

Round 1

Reviewer 1 Report

Comments and Suggestions for Authors

The manuscript "Patterns of spatial variation in rumen microbiology, histomorphology, and fermentation parameters in Tarim wapiti (Cervus elaphus yarkandensis)" is written on the basis of an original study and contains a novel and significant data on the biology of this deer.

However, there are some unclarities, which should be addressed before publication of the manuscript.

1) Line 138-163. The protocol of molecular procedures described here needs thorough check-up. For instance, in line 149-150 the authors have written that "PCR products were mixed in equidensity ratios." First, PCR products all have absolutely equal density. But they differed in molar and ng/mL concentrations. Then, mixing of PCR products is usually used when PCR with every sample DNA is performed in replicas, or just before sequencing. In the latter case this procedure is called pooling of DNA libraries. But, in this manuscript neither PCR replicas no DNA libraries pooling are not mentioned at all. Next example: NEB Next® Ultra™ II FS DNA PCR-free Library Prep Kit is not used for DNA libraries preparation from PCR products, but from genomic or total DNA and is inadmissible for DNA metabarcoding performed in this manuscript. It is even reflected in the kit's name: "PCR-free Library Prep Kit". I strongly recommend the authors to request an exact protocol of DNA isolation, DNA libraries preparation and their sequencing in the company that have performed sequencing for this study.

2) ASVs are indicated in Line 167, but in the rest text OTUs term is used everywhere. This contradiction should be removed, only ASV or OTU can be operated due to different algorithms of their producing.

3) Line 214. "bray distance" is not correct, the correct name is "Bray-Curtis distance".

4) A contradiction between the abstract ("Acetobacter had a significant positive association with pH") and lines 267-268 ("pH being positively correlated with Verrucomicrobia and negatively correlated with Thermoproteota, Acetobacteraceae, and Acetobacter").

5) The abstract is written not well, because it does not reflect all main findings of this study. I recommend to correct the abstract according to the conclusion.

Comments on the Quality of English Language

English needs thorough and extensive ediditing due to numerous grammar and style mistakes and improper phrases.

Author Response

Dear Editors and Reviewers:

We appreciate your invitation to revise our manuscript again and the valuable comments by the reviewers. The manuscript has been carefully revised according to the comments.

Reviewer: 1

General comments:

The manuscript "Patterns of spatial variation in rumen microbiology, histomorphology, and fermentation parameters in Tarim wapiti (Cervus elaphus yarkandensis)" is written on the basis of an original study and contains a novel and significant data on the biology of this deer.

However, there are some unclarities, which should be addressed before publication of the manuscript.

AU: Thank you for your nice compliments on our manuscript. We thoroughly rectified every mistake in the amended text, including grammatical and stylistic flaws and bad word choice, with the assistance of our colleagues. Thank you for your warm compliments and constructive critique on our work.

Line Comment

  1. Line 138-163. The protocol of molecular procedures described here needs thorough check-up. For instance, in line 149-150 the authors have written that "PCR products were mixed in equidensity ratios." First, PCR products all have absolutely equal density. But they differed in molar and ng/mL concentrations. Then, mixing of PCR products is usually used when PCR with every sample DNA is performed in replicas, or just before sequencing. In the latter case this procedure is called pooling of DNA libraries. But, in this manuscript neither PCR replicas no DNA libraries pooling are not mentioned at all. Next example: NEB Next® Ultra™ II FS DNA PCR-free Library Prep Kit is not used for DNA libraries preparation from PCR products, but from genomic or total DNA and is inadmissible for DNA metabarcoding performed in this manuscript. It is even reflected in the kit's name: "PCR-free Library Prep Kit". I strongly recommend the authors to request an exact protocol of DNA isolation, DNA libraries preparation and their sequencing in the company that have performed sequencing for this study.

AU: We have altered the presentation after requesting the full techniques for DNA isolation, DNA library preparation, and sequencing from the firm that sequenced it. Lines151-155, 157-162.

  1. ASVs are indicated in Line 167, but in the rest text OTUs term is used everywhere. This contradiction should be removed, only ASV or OTU can be operated due to different algorithms of their producing.

AU: We removed this discrepancy after confirming that this sequencing was utilized for ASV and described as ASV in the unified paper. Lines172, 173, 178, 217, 218.

  1. Line 214. "bray distance" is not correct, the correct name is "Bray-Curtis distance".

AU: Throughout the text, we have updated the name "bray distance" to the right name "Bray-Curtis distance". Lines 180, 220.

  1. A contradiction between the abstract ("Acetobacter had a significant positive association with pH") and lines 267-268 ("pH being positively correlated with Verrucomicrobia and negatively correlated with Thermoproteota, Acetobacteraceae, and Acetobacter").

AU: We reviewed the data and resolved the contradiction here.

  1. The abstract is written not well, because it does not reflect all main findings of this study. I recommend to correct the abstract according to the conclusion.

AU: We have revised the abstract considering our discoveries. Lines 12-31.

  1. Comments on the Quality of English Language: English needs thorough and extensive ediditing due to numerous grammar and style mistakes and improper phrases.

AU: We used the help of colleagues to rectify grammatical and stylistic problems, as well as poor word choice, in the manuscript.

Reviewer 2 Report

Comments and Suggestions for Authors

Even though the study has an interesting approach, the authors failed to critically scrutinize their statements and check the plausibility of the statements and data.

Line 29: "The study showed that Prevotellaceae had a positive relationship with Propionate". Line 270: "Propionate content was positively correlated with Prevotellaceae".Table 3.

According to Table 3, the propionate concentration fluctuated between 11.78 and 11.32 mmol/l with an SEM of 0.14, i.e. less than 5%. How can a significant correlation be claimed with such a small fluctuation range?  The fluctuations must also be seen against the background of the measurement accuracy.

Line 29: "Acetobacteraceae and its genus Acetobacter had a significant positive association with pH".

Line 267 : "with pH being positively correlated with Verrucomicrobia and negatively correlated with Thermoproteota"

The pH value fluctuated between 5.92 and 5.97. How can a significant correlation be claimed with such a small fluctuation range?  The fluctuations must also be seen against the background of the measurement accuracy.

Line 102:

Unfortunately, the authors do not describe how they ensured that no mixing of the rumen contents between the individual regions took place after slaughter. If separation is already difficult in the living animal due to motility, it is almost impossible post mortem. The question must therefore be asked whether separate segments of the rumen contents were actually examined.

Line 113

It is not stated at how many points of the sample or at how many samples the histomorphic values were recorded. There is also no figure showing which reference points or lines the authors used. How were villus sections without contact to the submucosa handled? How was the different thickness of the musculature (see Fig. 2f) recorded?

There is also no clear description of how the uniform positioning of the samples was ensured. How was excessive proximity to the rumen pillars and thus a falsification of the results avoided?

 Tables

The n-data are missing in all tables. It therefore remains unclear whether the statistics were based on the technical or biological replicates.

 Minor points

Line 19: SEF is not explained

Line 29: "relationship with propionate" It is unclear whether the propionate concentration, the production or the disappearance rate is meant.

Line 125: R0 is not explained

Line 270: "Propionate content was positively correlated with Prevotellaceae". The authors did not determine the content but the concentration.

Table2: In some cases 6 digits are given. Under no circumstances can it be assumed that the analysis had this measurement accuracy.

Table3: Mmol/L is not a comprehensible specification of a unit (M = molarity, m=milli)

Author Response

Reviewer: 2

Comments:

Even though the study has an interesting approach, the authors failed to critically scrutinize their statements and check the plausibility of the statements and data.

AU: Thank you for your comments. We assessed the manuscript's presentation critically, verified the presentation and data for accuracy, and revised the text. Thank you for your review and remarks once more.

Line Comment

  1. Line 29: "The study showed that Prevotellaceae had a positive relationship with Propionate". Line 270: "Propionate content was positively correlated with Prevotellaceae". Table 3. According to Table 3, the propionate concentration fluctuated between 11.78 and 11.32 mmol/l with an SEM of 0.14, i.e. less than 5%. How can a significant correlation be claimed with such a small fluctuation range? The fluctuations must also be seen against the background of the measurement accuracy.

AU: We precisely assessed the data and found that the Prevotellaceae is positively connected with propionate content; however, the SEM for propionate content is 0.14, but the SEM for Prevotellaceae abundance is 0.75, i.e. larger than 5%. Table 6.

  1. Line 29: "Acetobacteraceae and its genus Acetobacter had a significant positive association with pH".

AU: We reviewed the data and resolved the contradiction here.

  1. Line 267 : "with pH being positively correlated with Verrucomicrobia and negatively correlated with Thermoproteota". The pH value fluctuated between 5.92 and 5.97. How can a significant correlation be claimed with such a small fluctuation range? The fluctuations must also be seen against the background of the measurement accuracy.

AU: We reanalyzed the manuscript for correlation analysis by recounting the data and removing microorganisms with low abundance. Figure 5.

  1. Line 102: Unfortunately, the authors do not describe how they ensured that no mixing of the rumen contents between the individual regions took place after slaughter. If separation is already difficult in the living animal due to motility, it is almost impossible post mortem. The question must therefore be asked whether separate segments of the rumen contents were actually examined.

AU: We have added the process of collecting rumen contents from each region after slaughter as slightly as possible to avoid causing mixing. Lines 100-106.

  1. Line 113

It is not stated at how many points of the sample or at how many samples the histomorphic values were recorded. There is also no figure showing which reference points or lines the authors used. How were villus sections without contact to the submucosa handled? How was the different thickness of the musculature (see Fig. 2f) recorded?

AU: We've provided an explanation of the number of samples used to capture histomorphometric results as well as a reference line. Lines123-126, figure 2.

  1. There is also no clear description of how the uniform positioning of the samples was ensured. How was excessive proximity to the rumen pillars and thus a falsification of the results avoided?

AU: We have added descriptions of the various regions sampled, located in the center of each region. Lines 109-110.

  1. Tables

The n-data are missing in all tables. It therefore remains unclear whether the statistics were based on the technical or biological replicates.

AU: We supplied n-data to all tables.

  1. Line 19: SEF is not explained

AU: The explanation of the SEF has been inserted in line 127.

  1. Line 29: "relationship with propionate" It is unclear whether the propionate concentration, the production or the disappearance rate is meant.

AU: We have revised to propionate content and reviewed the manuscript.

  1. Line 270: "Propionate content was positively correlated with Prevotellaceae". The authors did not determine the content but the concentration.

AU: We have revised to propionate content and reviewed the manuscript.

  1. Table2: In some cases 6 digits are given. Under no circumstances can it be assumed that the analysis had this measurement accuracy.

AU: We converted um to mm for statistical analysis, and the table's data expression has been changed. Table2.

  1. Table3: Mmol/L is not a comprehensible specification of a unit (M = molarity, m=milli)

AU: We have corrected the units of VFAs. Table 3.

Round 2

Reviewer 1 Report

Comments and Suggestions for Authors

The authors have performed a great work to address the unclarities indicated in my review. However, some unclarities remain in the manuscript without changes. They are indicated below and should be addressed before publication. Otherwise, this study will be irreproducible.

1) Line 153. "PCR products were mixed in equidensity ratios." This phrase is incorrect and irrelevant here.

First, PCR products all have absolutely equal density. But they differed in molar and ng/mL concentrations. So, term "equidensity" is incorrect, correct terms are "equimolar" or "in equal concentrations".

Second, mixing of PCR products at this stage results fail NGS run due to impossibility to differentiate samples, because at this stage individual indices that are specific for every sample have not been added yet to the amplicons. So, this sentence should be replace from here. There are two points in protocol of DNA libraries preparing. First, mixing of PCR products is usually used when the first PCR is performed in replicas for every sample. In such case number of replicas should be indicated, e.g. "PCR products from three replicas were mixed for every sample". Also, mixing of PCR products is used before sequencing for pooling the DNA libraries. In this case the phrase mentioned above shoud be replaced just before DNA libraries pooling. 

2) Lines 157-158. "Sequencing libraries were generated using NEB Next®Ultra™ II FS DNA PCR-free Library Prep Kit (New England Biolabs, USA, Catalog #: E7430L) following manufacturer's recommendations and indexes were added." Unfortunately, this kit is incompactible with 16S rDNA libraries, and is not used for DNA libraries preparation from PCR products. It is even reflected in the kit's name: "PCR-free Library Prep Kit". In the manual for this kit, pages 2 and 5 (the file is attached) it is directlly indicated that starting material is genomic DNA only.

Unclarities described above make this study iireproductive and their addressing is necessary before publishing the manuscript. So, I strongly recommend the authors to request a detailed protocol of DNA libraries preparation and their sequencing in the company that have performed sequencing for this study, and then to correct a description of the protocol in the manuscript.

3) According to the International Code of Nomenclature of Prokaryotes, 2008 Revision (published in IJSEM, Volume 69, issue 1A, on 11 January 2019) Chapter 4, Part A all the Latin names of Prokaryotes should be indicated "by a different type face, e.g., italic, or by some other device to distinguish them from the rest of the text.". So it would be correct and relevant to write all Latin names of bacterial taxa in the manuscript, from phylum to species, using Italic font.

In addition, I have tried to correct some improper phrases and grammar mistakes remained in the manuscript and the authors can see my corrections in comments inside the PDF file attached.

Comments on the Quality of English Language

The authors have corrected a lot of mistakes and improper phrases. I also have tried to correct some improper phrases and grammar mistakes remained in the manuscript and the authors can see my corrections in comments inside the PDF file attached. However, English proofreading is desirable.

Author Response

Reviewer: 1

The authors have performed a great work to address the unclarities indicated in my review. However, some unclarities remain in the manuscript without changes. They are indicated below and should be addressed before publication. Otherwise, this study will be irreproducible.

AU: Thank you for the positive comments and beneficial adjustments on our revised manuscript. Each problem has been thoroughly rectified in accordance with your proposed adjustments. Thank you for your kind words and constructive criticism on our work.

Line Comment

  1. Line 153. "PCR products were mixed in equidensity ratios." This phrase is incorrect and irrelevant here.

First, PCR products all have absolutely equal density. But they differed in molar and ng/mL concentrations. So, term "equidensity" is incorrect, correct terms are "equimolar" or "in equal concentrations".

AU: We have modified it to be correct, to equimolar. Lines157.

Second, mixing of PCR products at this stage results fail NGS run due to impossibility to differentiate samples, because at this stage individual indices that are specific for every sample have not been added yet to the amplicons. So, this sentence should be replace from here. There are two points in protocol of DNA libraries preparing. First, mixing of PCR products is usually used when the first PCR is performed in replicas for every sample. In such case number of replicas should be indicated, e.g. "PCR products from three replicas were mixed for every sample". Also, mixing of PCR products is used before sequencing for pooling the DNA libraries. In this case the phrase mentioned above should be replaced just before DNA libraries pooling.

AU: We amplify genomic nucleic acids using a barcode so that the amplified products may be recognized when combined. During our research, each nucleic acid is amplified as a single sample with a unique barcode, and we do not discriminate between duplicate parallel samples when combining samples for library development. Lines 151-155.

  1. Lines 157-158. "Sequencing libraries were generated using NEB Next®Ultra™ II FS DNA PCR-free Library Prep Kit (New England Biolabs, USA, Catalog #: E7430L) following manufacturer's recommendations and indexes were added." Unfortunately, this kit is incompatible with 16S rDNA libraries, and is not used for DNA libraries preparation from PCR products. It is even reflected in the kit's name: "PCR-free Library Prep Kit". In the manual for this kit, pages 2 and 5 (the file is attached) it is directlly indicated that starting material is genomic DNA only.

AU: Regarding the library construction kit, we reviewed the instructions, and PCR-free meaning that no amplification is conducted during the library building process, rather than needing no amplification prior to library creation. Furthermore, the kit instructions do not state that the amplified DNA sequence cannot be utilized.

Unclarities described above make this study iireproductive and their addressing is necessary before publishing the manuscript. So, I strongly recommend the authors to request a detailed protocol of DNA libraries preparation and their sequencing in the company that have performed sequencing for this study, and then to correct a description of the protocol in the manuscript.

AU: We engaged with sequencing businesses about modifying DNA library preparation and sequencing techniques.

  1. According to the International Code of Nomenclature of Prokaryotes, 2008 Revision (published in IJSEM, Volume 69, issue 1A, on 11 January 2019) Chapter 4, Part A all the Latin names of Prokaryotes should be indicated "by a different type face, e.g., italic, or by some other device to distinguish them from the rest of the text.". So it would be correct and relevant to write all Latin names of bacterial taxa in the manuscript, from phylum to species, using Italic font.

AU: We italicized all bacterial taxa in the manuscript, from phylum to species.

  1. In addition, I have tried to correct some improper phrases and grammar mistakes remained in the manuscript and the authors can see my corrections in comments inside the PDF file attached.

AU: Thank you very much for your adjustments to the improper phrases and grammar mistakes in the manuscript; we have revised each remark based on the comments in the attached PDF. Additionally, modifications to the manuscript's grammar.

  1. The authors have corrected a lot of mistakes and improper phrases. I also have tried to correct some improper phrases and grammar mistakes remained in the manuscript and the authors can see my corrections in comments inside the PDF file attached. However, English proofreading is desirable.

AU: Thank you very much for your adjustments to the improper phrases and grammar mistakes in the manuscript; we have revised each remark based on the comments in the attached PDF. Additionally, modifications to the manuscript's grammar.

Reviewer 2 Report

Comments and Suggestions for Authors

The corrections have not improved the manuscript. My comments are included in the authors' response: "REV"

Even though the study has an interesting approach, the authors failed to critically scrutinize their statements and check the plausibility of the statements and data.

AU: Thank you for your comments. We assessed the manuscript's presentation critically, verified the presentation and data for accuracy, and revised the text. Thank you for your review and remarks once more.

Line Comment

  1. Line 29: "The study showed that Prevotellaceae had a positive relationship with Propionate". Line 270: "Propionate content was positively correlated with Prevotellaceae". Table 3. According to Table 3, the propionate concentration fluctuated between 11.78 and 11.32 mmol/l with an SEM of 0.14, i.e. less than 5%. How can a significant correlation be claimed with such a small fluctuation range? The fluctuations must also be seen against the background of the measurement accuracy.

AU: We precisely assessed the data and found that the Prevotellaceae is positively connected with propionate content; however, the SEM for propionate content is 0.14, but the SEM for Prevotellaceae abundance is 0.75, i.e. larger than 5%. Table 6.

REV: If a parameter fluctuates by less than 5%, then with n=6 a correlation is a purely random finding. But even if the correlation actually exists, it is biologically irrelevant.

  1. Line 29: "Acetobacteraceae and its genus Acetobacter had a significant positive association with pH".

AU: We reviewed the data and resolved the contradiction here.

  1. Line 267 : "with pH being positively correlated with Verrucomicrobia and negatively correlated with Thermoproteota". The pH value fluctuated between 5.92 and 5.97. How can a significant correlation be claimed with such a small fluctuation range? The fluctuations must also be seen against the background of the measurement accuracy.

AU: We reanalyzed the manuscript for correlation analysis by recounting the data and removing microorganisms with low abundance. Figure 5.

REV: See my comments above

  1. Line 102: Unfortunately, the authors do not describe how they ensured that no mixing of the rumen contents between the individual regions took place after slaughter. If separation is already difficult in the living animal due to motility, it is almost impossible post mortem. The question must therefore be asked whether separate segments of the rumen contents were actually examined.

AU: We have added the process of collecting rumen contents from each region after slaughter as slightly as possible to avoid causing mixing. Lines 100-106.

REV.: I could not find any description of a method that would have prevented the mixing of the different compartments. It is also completely unclear how the different compartments were specifically targeted for sampling.  

  1. Line 113

It is not stated at how many points of the sample or at how many samples the histomorphic values were recorded. There is also no figure showing which reference points or lines the authors used. How were villus sections without contact to the submucosa handled? How was the different thickness of the musculature (see Fig. 2f) recorded?

AU: We've provided an explanation of the number of samples used to capture histomorphometric results as well as a reference line. Lines123-126, figure 2.

 REV: No reference lines for measuring the rumen villi are shown. It is not shown how the localizations were randomized for the measurement of the rumen villi. If "more typical sites were chosen" then the selection and evaluation is falsified from the outset.  Numerous randomly selected locations are also necessary for measuring the muscle layer.  

AU: We have added descriptions of the various regions sampled, located in the center of each region. Lines 109-110.

  1. Line 29: "relationship with propionate" It is unclear whether the propionate concentration, the production or the disappearance rate is meant.

AU: We have revised to propionate content and reviewed the manuscript.

  1. Line 270: "Propionate content was positively correlated with Prevotellaceae". The authors did not determine the content but the concentration.

AU: We have revised to propionate content and reviewed the manuscript.

REV: Why this way around?

  1. Table3: Mmol/L is not a comprehensible specification of a unit (M = molarity, m=milli)

AU: We have corrected the units of VFAs. Table 3.

REV: mMol/L is also an unauthorized designation.: https://en.wikipedia.org/wiki/Molar_concentration

Author Response

Reviewer: 2

Comments:

Even though the study has an interesting approach, the authors failed to critically scrutinize their statements and check the plausibility of the statements and data.

AU: Thank you for your comments. We assessed the manuscript's presentation critically, verified the presentation and data for accuracy, and revised the text. Thank you for your review and remarks once more.

Line Comment

  1. Line 29: "The study showed that Prevotellaceae had a positive relationship with Propionate". Line 270: "Propionate content was positively correlated with Prevotellaceae". Table 3. According to Table 3, the propionate concentration fluctuated between 11.78 and 11.32 mmol/l with an SEM of 0.14, i.e. less than 5%. How can a significant correlation be claimed with such a small fluctuation range? The fluctuations must also be seen against the background of the measurement accuracy.

AU: We precisely assessed the data and found that the Prevotellaceae is positively connected with propionate content; however, the SEM for propionate content is 0.14, but the SEM for Prevotellaceae abundance is 0.75, i.e. larger than 5%. Table 6.

REV: If a parameter fluctuates by less than 5%, then with n=6 a correlation is a purely random finding. But even if the correlation actually exists, it is biologically irrelevant.

AU: We reanalyzed the data, taking your comments into consideration, and together with these criteria, we deleted the correlation analysis for the purpose of data rigor and dependability, as well as later reference value.

  1. Line 267 : "with pH being positively correlated with Verrucomicrobia and negatively correlated with Thermoproteota". The pH value fluctuated between 5.92 and 5.97. How can a significant correlation be claimed with such a small fluctuation range? The fluctuations must also be seen against the background of the measurement accuracy.

AU: We reanalyzed the manuscript for correlation analysis by recounting the data and removing microorganisms with low abundance. Figure 5.

REV: See my comments above

AU: We reanalyzed the data, taking your comments into consideration, and together with these criteria, we deleted the correlation analysis for the purpose of data rigor and dependability, as well as later reference value.

  1. Line 102: Unfortunately, the authors do not describe how they ensured that no mixing of the rumen contents between the individual regions took place after slaughter. If separation is already difficult in the living animal due to motility, it is almost impossible post mortem. The question must therefore be asked whether separate segments of the rumen contents were actually examined.

AU: We have added the process of collecting rumen contents from each region after slaughter as slightly as possible to avoid causing mixing. Lines 100-106.

REV.: I could not find any description of a method that would have prevented the mixing of the different compartments. It is also completely unclear how the different compartments were specifically targeted for sampling.

AU: We have provided details of methods to prevent compartment mixing and how to specialize sampling for individual compartments. Lines97-104.

  1. Line 113

It is not stated at how many points of the sample or at how many samples the histomorphic values were recorded. There is also no figure showing which reference points or lines the authors used. How were villus sections without contact to the submucosa handled? How was the different thickness of the musculature (see Fig. 2f) recorded?

AU: We've provided an explanation of the number of samples used to capture histomorphometric results as well as a reference line. Lines123-126, figure 2.

REV: No reference lines for measuring the rumen villi are shown. It is not shown how the localizations were randomized for the measurement of the rumen villi. If "more typical sites were chosen" then the selection and evaluation is falsified from the outset.  Numerous randomly selected locations are also necessary for measuring the muscle layer.

AU: We defined a reference line for rumen villi measurement and explained the randomization of rumen villi and muscle layer measurement. Figure2, Lines123-126.

  1. Line 270: "Propionate content was positively correlated with Prevotellaceae". The authors did not determine the content but the concentration.

AU: We have revised to propionate content and reviewed the manuscript.

REV: Why this way around?

AU: We began by measuring the propionate content, which was mistakenly confused at the time when it was written but has since been corrected.

  1. Table3: Mmol/L is not a comprehensible specification of a unit (M = molarity, m=milli)

AU: We have corrected the units of VFAs. Table 3.

REV: mMol/L is also an unauthorized designation.: https://en.wikipedia.org/wiki/Molar_concentration

AU: We have corrected the units of VFAs. Table 3.